# Effect of Gravity on Bacterial Adhesion to Heterogeneous Surfaces

**DOI:** 10.3390/pathogens12070941

**Published:** 2023-07-15

**Authors:** Kayla Hogan, Sai Paul, Guanyou Lin, Jay Fuerte-Stone, Evgeni V. Sokurenko, Wendy E. Thomas

**Affiliations:** 1Department of Bioengineering, University of Washington, Seattle, WA 98195, USA; 2Department of Microbiology, University of Washington, Seattle, WA 98195, USA; evs@uw.edu

**Keywords:** bacteria, adhesion, hydrodynamics, *Escherichia coli*

## Abstract

Bacterial adhesion is the first step in the formation of surface biofilms. The number of bacteria that bind to a surface from the solution depends on how many bacteria can reach the surface (bacterial transport) and the strength of interactions between bacterial adhesins and surface receptors (adhesivity). By using microfluidic channels and video microscopy as well as computational simulations, we investigated how the interplay between bacterial transport and adhesivity affects the number of the common human pathogen *Escherichia coli* that bind to heterogeneous surfaces with different receptor densities. We determined that gravitational sedimentation causes bacteria to concentrate at the lower surface over time as fluid moves over a non-adhesive region, so bacteria preferentially adhere to adhesive regions on the lower, inflow-proximal areas that are downstream of non-adhesive regions within the entered compartments. Also, initial bacterial attachment to an adhesive region of a heterogeneous lower surface may be inhibited by shear due to mass transport effects alone rather than shear forces per se, because higher shear washes out the sedimented bacteria. We also provide a conceptual framework and theory that predict the impact of sedimentation on adhesion between and within adhesive regions in flow, where bacteria would likely bind both in vitro and in vivo, and how to normalize the bacterial binding level under experimental set-ups based on the flow compartment configuration.

## 1. Introduction

As bacterial resistance to antibiotics has increased [1,2,3,4,5], researchers have sought alternative methods of preventing bacterial infections. A relatively new approach is to prevent bacterial adhesion to and thus biofilm formation on host cell or tissue surfaces or devices such as urinary or blood catheters, joint implants, tracheal tubing, etc. [6,7]. Bacterial adhesion is mediated by adhesins that recognize specific carbohydrate or protein receptors. The function of adhesins can be blocked by small-molecule inhibitors or specific antibodies [7,8]. Alternatively, the surfaces of implanted devices can be modified to minimize the ability of bacteria to adhere to them [7,9]. Nearly universally, bacterial pathogens adhere to surfaces in the presence of fluid flow, and the receptors to which they bind are not uniformly distributed. Thus, it is important to understand how bacterial adhesion is affected by fluid flow and surface heterogeneity.

Fluid flow-derived shear stress generates drag force on a bound bacterium. While some bacteria show shear-inhibited adhesion, in which they attach in higher numbers at lower shear stress [10,11,12,13,14,15,16,17,18,19,20,21,22,23,24,25,26,27], many bind in a shear-enhanced manner, in which they attach in higher numbers at higher shear stress [14,28,29]. The amount of shear stress needed to inhibit adhesion has been used to estimate or compare the strength of adhesive interactions [11,22,25,30,31]. Moreover, the effect of adhesion inhibitors depends on shear stress [17,26,32,33], and understanding the mechanism underlying shear-enhanced adhesion has been critical for designing more effective inhibitors [34,35,36]. This has been especially true for *Escherichia coli*, one of the most common human pathogens, which causes a majority of diarrhea cases and urinary tract infections (UTIs) (including Foley catheter-associated UTIs) [37,38,39], as well as bloodstream infections [40,41], pneumonia in mechanically ventilated patients [42], and other extra-intestinal infections [43]. *E. coli* were the first bacteria for which flow-dependent shear-enhanced adhesion was demonstrated and its molecular mechanism was studied in great detail [7,29,44,45,46,47]. The most common adhesin of *E. coli*, the type 1 fimbrial adhesin FimH, has been a target for antiadhesive therapy to prevent urinary tract infections [7,47], and mediates shear-enhanced adhesion [29,44,45,46]. FimH binds mannosylated receptors via a catch bond mechanism, where the strength of binding to mannose is allosterically enhanced by the application of tensile force that induces separation between the adhesive and fimbria-anchoring domains of FimH *E. coli* [7,37].

Since the discovery of shear-enhanced adhesion in *E. coli*, this phenomenon has been demonstrated in over a dozen different bacterial species [18,28,48,49,50,51,52,53,54,55,56,57,58,59], and studies of bacterial adhesion in flow chambers or other microfluidic devices are now regularly performed [60,61,62,63,64]. However, to interpret measurements of bacterial adhesion to a surface in flow, it is critical to determine how adhesion reflects the intrinsic strength of binding versus the number of bacteria available near the surface as a result of mass transport. The latter depends on advection (the movement of particles with fluid flow, sometimes referred to as convection), diffusion (the movement of particles through fluid towards regions of lower chemical potential), the attachment of particles in fluid to the surface, and gravitational sedimentation. For very small particles like macromolecular complexes or viruses, the effect of gravity on their sedimentation can be neglected [65,66,67], but this is not the case for bacteria [63,68] or larger eukaryotic cells [30,69]. One study by Li et al. predicted that the transport-limited rate of bacterial attachment to a lower surface in flow is equal to the product of the sedimentation velocity and the bulk concentration [63]. This theory was validated by correctly predicting the rate of adhesion to uniform abiotic lower surfaces, especially at low flow, where drag force is unlikely to impact adhesive strength [63].

However, the theory by Li et al. may not be appropriate for other situations in which bacteria adhere in flow to highly heterogeneous surfaces due to the localized expression of receptors on the target cells and tissues. Moreover, in vitro studies of bacterial adhesion to biological receptors often use a microfluidic device in which the receptors are deposited in a single large droplet [29,70] or multiple tiny droplets to create a microarray [71,72,73]. Mass transport effects have been addressed for the adhesion of macromolecules in these situations [67]. However, it cannot be assumed that these conclusions will be relevant for particles like bacteria that can sediment under gravitational forces, and the few studies that have characterized the adhesion of whole bacteria to arrayed compounds have not addressed the impact of mass transport [74,75]. It is thus relevant to understand how transport affects bacterial adhesion to heterogeneous surfaces.

Here we seek to understand and predict how diffusion, gravitational sedimentation, and fluid advection all contribute to bacterial transport and thus to adhesion on heterogeneous lower surfaces. For that, we used as a model the adhesion of type 1 fimbriated *E. coli* to the lower surface of the flow chamber, in which only a small region is coated with mannosylated glycoproteins. We employed both video microscopy and computational transport simulations to analyze bacterial binding.

## 2. Materials and Methods

Bacterial culture: The pPKL114 and pGB2-24 plasmids were transformed into DH5α *Escherichia coli* bacteria, so that the bacteria expressed the type 1 fimbriae with tip associated mannose-binding FimH adhesin variant of E. coli strain K12 (or J96), broadly used for the structural, functional, immunization, and small molecule inhibitory studies of FimH [29]. These bacteria were grown overnight in Lennox Lysogeny Broth with 100 µg/mL Ampicillin and 30 µg/mL Chloramphenicol, rinsed twice with Phosphate Buffered Saline (PBS) and then diluted to an optical density of 0.03–0.05 at 600 nm in PBS with 0.1% bovine serum albumin (PBS-BSA).

Preparation of the flow chamber’s lower surfaces. Bovine RNAseB, which is rich in high-mannose-type oligosaccharides [76] recognized by type 1 fimbriae under both static and flow conditions, was incubated with Corning^TM^ 35 mm tissue culture dishes. For small adhesive spots, 0.1 mg/mL RNAseB in bicarbonate buffer pH 8.0 was drawn in a line with a pipette tip at the indicated location. For large adhesive spots, lines were drawn in the same manner, and then a 60 µL droplet of the same RNAseB solution was deposited over an area of approximately 0.5 by 0.75 cm to one side of the line. Dishes were incubated at 37 °C for 75 min, rinsed three times with PBS, and incubated overnight in PBS BSA at 4 °C to block nonspecific adhesion. To prepare nonadhesive lower surfaces for controls and for measuring the concentration of bacteria flowing near the lower surface, Corning^TM^ 35 mm tissue culture dishes were simply blocked with PBS-BSA overnight at 4 °C.

Bacterial adhesion in flow. The mannosylated dishes were installed as the lower surface of a Glycotech^TM^ flow chamber with the “B” gasket, creating a channel that is 2 cm long, 0.25 cm wide, and 254 µm high. Bacteria suspended in PBS-BSA at an optical density of 0.03 to 0.05 were washed through the flow chamber at a high shear rate of 3100 s^−1^ for 5 to 10 s to introduce the bacteria into the chamber, and then washed at 15 s^−1^ for 35 min while the mannosylated lower surface at the indicated locations was imaged every 30 s in a Nikon TE2000 inverted microscope with a CoolSnap CCD camera, and Micro-Manager software. The camera shutter speed was set to 634 ms to blur out the unbound bacteria [29]. To measure only stationary-bound bacteria, two consecutive images were binarized and multiplied together before counting with the ImageJ particle counter.

Measurement of the near-surface concentration in flow. To measure the concentration of bacteria near the lower surface in flow, nonadhesive surfaces were prepared as described above and installed into a Glycotech flow chamber. Bacteria were infused at a shear rate of 4.7 s^−1^, and images were collected with a 0.1-s framerate at the indicated times with the lower surface in focus. The moving bacteria were counted after thresholding and binarizing the images using ‘Analyze Particles’ in ImageJ, with a range of 8 to 50 pixels^2^, to allow for elongated images due to bacteria movement. The near-surface concentration of bacteria was then measured by dividing the number counted by the area of the images and the depth of the measured field, which, in turn, was calculated to be 2.5 µm using the formula field depth = maximum bacterial velocity/shear rate. (Bacteria farther from the surface were faded and blurred because they moved at too high a velocity and/or were out of focus.)

Measurement of diffusion coefficient D and Sedimentation Velocity $V_{g}$: A chamber was built using two glass slides separated by two rows of double-sided tape and blocked overnight at 4 °C with PBS-BSA. The bacteria were pipetted into the chamber and allowed to settle to one surface of the chamber, at which point the chamber was flipped over and a phase contrast microscope with a CCD camera was used to image the bacteria every 2 s as they settled. The depth of the chamber was determined by the distance the objective was moved between focusing on the top and the bottom of the chamber, as previously described [77], and determined to be 59 µm. The sedimentation velocity of each bacterium was determined by dividing the chamber height by the time taken for the bacterium to settle into focus on the lower surface, as shown in Appendix A. The mean was found to be $V_{g}=0.108\;\mu m/s\;$with a standard deviation of 0.037 μm/s, and standard error of the mean of 0.0047 (n = 62). The settled bacteria were then imaged at 10 images per second and the three-dimensional locations of each bacteria were identified with a previously described tracking algorithm [77]. The diffusion coefficient of each bacterium was calculated as
(1)$$D=\left(x^{2}+y^{2}\right)/4t$$
where *x* and *y* indicate the distance traveled in the *x* and *y* directions over time *t*. The mean diffusion coefficient was $D=0.19\;\pm \;0.03\;{\mu m}^{2}/s\;$(n = 22).

Computational Model Development. A COMSOL two-dimensional model was built to mimic the Glychotech^TM^ flow chamber used in experiments. The chamber was represented by a 2 cm long by 254 µm high rectangle. To model experiments with a single large spot, the adhesive spot started 10 mm from the inlet of the chamber and was 0.5 mm long. The velocity of bacteria in the y direction was the sedimentation velocity, $v_{y}=-V_{g}$. The velocity of bacteria in the x-direction was that of parabolic fluid flow:(2)$$v_{x}=S\cdot \left(h-\frac{h^{2}}{H}\right)$$
where *S* is the wall shear rate (s^−1^), *h* is the distance between the bacterial midpoint and the lower surface, and *H* is the depth of the chamber. Because bacteria have a finite, not infinitesimal, size, we needed to address the fact that bacteria can adhere to the lower surface when their midpoint is a distance of one radius (*r*) from the lower surface, but their velocity is approximately the velocity of fluid at their midpoint, not their bottom. We therefore defined y=h−r as the distance between the bottom of the bacterium and the lower surface as the independent variable in the simulations. That is,
(3)$$v_{x}=S\cdot \left(y+r-\frac{{\left(y+r\right)}^{2}}{H}\right)=S\cdot \left(y+r-\frac{y^{2}}{H}-2\frac{ry}{H}-\frac{r^{2}}{H}\right).$$

Because r≪H, $\frac{r^{2}}{H}\ll r$, and $\frac{ry}{H}\ll y$, these two terms can be neglected, resulting in the following velocity vector for the bacteria:(4)$$\vec{v}=\left(\;S\cdot \left(y+r-\frac{y^{2}}{H}\right),-V_{g}\right).$$

The transport of diluted species module was used to solve the two-dimensional advection–diffusion equation:(5)$$\frac{\partial C}{\partial t}=D\nabla^{2}C-\nabla \cdot C\vec{v}.$$

All boundaries were modeled with no flux conditions except the inflow and outflow, which were modeled as such, and the lower surface, which was modeled as non-saturating and irreversible adhesion by a simple flux out of the boundary at rate $-k_{on}C.$ The accumulated bacteria were reciprocally modeled with the lower dimension variable *B* defined only on the boundary, with the equation
(6)$$\frac{\partial B}{\partial t}=k_{on}C.$$

Here, $k_{on}$ is the effective bacterial association rate constant, which is the ratio of the adhesive flux *J* (in $m^{-2}s^{-1}$) to the concentration near the surface available to bind (in $m^{-3}$) and thus has units of m/s or length per time. Parameter values from Table 1 were used unless otherwise indicated. For the simulations with a nonadhesive surface, $k_{on}=0$ and the near-surface concentration was calculated by averaging the concentration of bacteria within 2.5 µm of the lower surface.

## 3. Results

### 3.1. Gravitational Sedimentation Affects the Concentration of Bacteria near the Lower Surface

In order to determine how mass transport with gravitational sedimentation affects the number of type 1 fimbriated *E. coli* available to bind to the lower surface in flow, the number of bacteria was monitored via video microscopy near the lower surface of a microfluidic chamber with a nonadhesive service (i.e., not covered with mannosylated glycoproteins). The monitoring was performed at three locations within the chamber located x = 0.5, 1, and 1.5 cm downstream from the inflow (Figure 1a). Bacteria within the bottom 2.5 µm of the chamber (hereafter called near-surface), were counted because bacteria farther from the surface were blurred as they moved at a higher velocity and/or were out of focus. We observed that the near-surface concentration of bacteria increased over time at the same rate at all three locations, until it reached an equilibrium value (Figure 1b), with the equilibrium value increasing approximately linearly with distance from the inflow (Figure 1c). This increase cannot be explained by diffusion, which would act to maintain the initial concentration of bacteria, but instead, these results suggest that gravitational sedimentation impacts the near-surface concentration of bacteria in a manner that depends on both time and position along the length of the chamber.

### 3.2. Effect of Mass Transport on the Concentration of Bacteria near a Lower Surface

We next used computational simulations to model the combined impact of fluid flow, gravity, and diffusion on the concentration of bacteria over time and position within a microfluidic chamber. We first measured the diffusion coefficient and sedimentation velocity of individual bacteria by tracking their movement through fluid in a static chamber, in order to determine the parameter values to use when comparing the simulations to experiments. The fluid velocity in the chamber is known from the chamber geometry, and the volumetric flow rate is set by a programmable pump.

The simulations (Figure 2a) showed that near the inflow, bacteria are at their original concentration (normalized to one in this image), but further into the chamber, there is an increased concentration of bacteria at the lower surface and depletion at the upper surface. To compare the simulations with the experiments, the concentration of bacteria in a 2.5 µm high volume nearest the surface (Figure 2a) was averaged at the same locations as in the experiments. These calculations reproduced the key experimental observations depicted in Figure 1 above. The bacterial concentration near the lower surface increased linearly over time at a similar rate at all spots until it reached an equilibrium concentration (Figure 2b), and the equilibrium concentration increased with distance from the inflow (Figure 2c). The simulations showed a small quantitative difference in the time to reach equilibrium compared to the experiments, but this is not unusual for simulations in which nearly all parameters are measured experimentally, and may be attributed to experimental error in the parameter measurements. Thus, the simulations reproduced the key aspects of sedimentation and can be useful for understanding the sedimentation behavior of bacteria in flow.

To explore how flowing through a non-adhesive region affects the concentration of bacteria available to bind to the lower surface at equilibrium, we performed mass transport simulations without any binding to the surfaces. The near-surface concentration at equilibrium increased approximately linearly with both position in the chamber (Figure 3a) and sedimentation velocity (Figure 3b). This confirms that the increase in near-surface concentration observed in the experiments is indeed due to gravitational sedimentation. The simulations also predicted that the near-surface concentration decreases with the diffusion coefficient (Figure 3c) and wall shear rate (Figure 3d). This is consistent with the idea that diffusion and lateral fluid flow mitigate sedimentation by, respectively, mixing or washing out the concentrated layer of bacteria near the lower surface. More specifically, we determined that the concentration of bacteria near the lower surface increased linearly with the distance from the inflow and with the square of the sedimentation velocity, but decreased inversely with both the diffusion coefficient and shear rate, at least within the range of values for which diffusion and sedimentation are both important (Appendix A).

Overall, these simulations demonstrate that the number of bacteria available to bind is dramatically affected by the transit of bacteria through a non-adhesive region within the flow compartment (in our case, a microfluidic channel). Specifically, there will be more bacteria available to bind as bacteria move further into a channel, and this effect would be most pronounced at low wall shear rates, and with larger bacteria, which settle more quickly and diffuse more slowly.

### 3.3. Effect of Mass Transport on Bacterial Adhesion

To consider the importance of gravitational sedimentation on bacterial adhesion in flow, we recognize that surfaces are likely to be heterogeneous, for example, because only some but not other cells express appropriate receptors in vivo. Thus, we performed experiments in which a spot on the lower surface of a chamber was functionalized with 100 μg/mL of bovine RNAseB, a model glycoprotein with high-mannose-type oligosaccharides to which type 1 fimbriated bacteria bind well at any shear. We prepared an adhesive spot 1 cm into the flow chamber, as illustrated in Figure 4a. *E. coli* were washed through these chambers at a wall shear rate of 15 s^−1^ for 29 min, and adherent bacteria were recorded using video microscopy with a field of view that included the first 500 µm from the upstream edge of the spot. The density of bound bacteria was highest at the upstream edge of the spot and dropped with distance, as illustrated in Figure 4a. The images were then divided into 50 µm wide regions in the direction of flow, to determine how the number of adherent bacteria depended on the distance into the spot. As seen in Figure 4b, the density of adherent bacteria was highest at the upstream edge of the spot proximal to the flow input and decayed within about 200 µm to a relatively constant value in the interior of the spot.

In order to better understand this adhesive pattern, transport simulations were performed with the inclusion of an adhesive process. Because in our experiments bacteria rarely moved or detached upon binding to the surface, and the bacterial concentration was too low for the surface to become saturated, we modeled adhesion as irreversible and independent of the bound density by using a single first-order effective association rate constant, $k_{on}$, which is the rate at which bacteria that are touching the surface bind to it irreversibly. After fitting this rate constant to the data, the simulations matched the experimental data closely (Figure 4b).

To describe the pattern of adhesion observed in both experiments and simulations, we noted that the data was well approximated by an exponential decay with a constant baseline (Figure 4b).
(7)$$B=\left(B_{edge}-B_{int}\right)\times \exp\left(-\frac{x}{x_{c}}\right)+B_{int}$$

That is, binding density (*B*) decayed exponentially with distance within the spot (*x*) over a characteristic “fall-off distance ($x_{c}$)” from a high density at the upstream edge of the spot ($B_{edge}$) to a position-independent interior density ($B_{int}$).

We next used the simulations to explore how various experimental conditions would affect the number of bound bacteria and fit the exponential Equation (7) to the resulting patterns of adhesion to describe the results. Notably, we observed that the fall-off distance $x_{c}$ was relatively insensitive to spot position, sedimentation velocity, and diffusion coefficient, and varied less than twofold over shear rates of 5 to 50 s^−1^ (Appendix A). Therefore, the impacts of experimental conditions on adhesion can best be analyzed by focusing on impacts on the edge density $B_{edge}$ and interior density $B_{int}$, as shown in Figure 4c.

The amount of adhesion at both the upstream edge and the interior of a spot increased linearly with its position within the chamber. This demonstrates that more bacteria will adhere to adhesive spots that are positioned farther downstream in a fluidic channel, relative to those farther upstream. Interestingly, however, the pattern of adhesion within a spot was relatively insensitive to position; there was little to no difference in the fall-off distance or in the ratio of edge to interior density. Binding within such a spot also increased strongly with sedimentation velocity and decreased as the diffusion coefficient increased. Because sedimentation velocity increases with the square of the bacterial radius and the diffusion coefficient decreases with radius, larger bacteria will adhere much more rapidly to adhesive spots in flow than smaller bacteria, all else being equal. Finally, the wall shear rate decreased the amount of binding. Not surprisingly, these effects of position, sedimentation, diffusion, and shear rate (Figure 4c) reflect the impacts of these variables on near-surface concentration at the lower surface (Figure 3) This suggests that the number of bacteria that adhere to flow to adhesive regions strongly reflects the sedimentation of bacteria while they travel over a non-adhesive lower surface.

We also asked a related question about the position of the spot, which is how an upstream adhesive spot affects bacterial adhesion to another adhesive spot downstream. To address this question, we added a second 0.5 mm wide adhesive spot 2 mm upstream in our simulations of bacterial transport and adhesion. As might be expected, fewer bacteria adhered to a spot at a given location when there was another adhesive spot upstream (Figure 5). However, the sharpness of the edge was similar with and without the preceding spot; both the fall-off distance and the ratio of edge to interior density appeared unchanged. We conclude that the total amount of adhesion measured within a spot can be affected not only by the position of the spot in question, but also by the heterogeneity of the lower surface upstream. At the same time, and quite remarkably, the distribution of bacteria binding within an adhesive spot appears to be independent of the presence of other adhesive spots upstream, just as it was independent of the position of the spot (Figure 4).

### 3.4. Sensitivity of Bacterial Adhesion to Binding Kinetics

Finally, we addressed how bacterial adhesion reflects the effective association rate constant $k_{on}$ by varying this parameter in simulations (Figure 6). Bacterial adhesion was independent of the association rate constant $k_{on}$ above a critical value. This is expected because when the association rate constant is high enough, binding is “transport-limited,” meaning that adhesion occurs so quickly that the limiting factor is how fast bacteria are brought to the lower surface. With lower association rate constants, when binding was not transport-limited, bacterial attachment increased with the association rate constant at both the upstream edge of the spot, and in the interior (Figure 6a), as might be expected. In addition, the pattern of adhesion changed with the association rate constant. For example, when the association rate constant increased from 0.4 to 2 µm/s, the upstream edge density increased 1.7-fold, while the interior density remained the same, so the ratio of the edge to interior density increased 1.7-fold (Figure 6a). Moreover, the fall-off distance decreased by a similar amount within the same range of association rate constants (Figure 6b). These findings predict that when binding is not transport-limited, stronger adhesion is reflected by a sharper upstream edge, in which adhesion drops by a higher ratio over a shorter distance, relative to weaker adhesion.

To test these predictions, we measured the pattern of bacterial adhesion in flow within a spot coated with either 100 µg/mL or 50 µg/mL of RNAseB located 1 cm into a chamber, and fit Equation (3) to the data (Figure 6c). There was more adhesion overall (*p* = 0.04 by two-way ANOVA) with 100 µg/mL than with 50 µg/mL RNAseB, so adhesion was not transport-limited in these experiments. We would therefore expect the upstream edge to be sharper on 100 µg/mL RNAseB than on 50 µg/mL RNAseB. Indeed, this is visually clear in Figure 6c, and from the quantification of the data using Equation (3): the ratio of edge to interior density decreased from 2.6 on 100 µg/mL RNAseB to 1.9 on 50 µg/mL RNAseB, while the fall-off distance increased from 40 µm to 128 µm (Figure 6c).

## 4. Discussion

In this study, we have asked how gravitational sedimentation impacts bacterial adhesion in flow, with a particular focus on heterogeneous surfaces, on which only relatively small areas support bacterial adhesion. We demonstrated that the concentration of bacteria near a non-adhesive lower surface increases over time until it reaches a plateau that increases with distance from the inflow. This finding was reproduced in simulations that also demonstrated that the concentration of bacteria near the lower surface decreases with fluidic shear rate. The simulations also showed that the rate of bacterial adhesion to a highly adhesive spot within an otherwise non-adhesive lower surface reflected the concentration near that surface. These findings predict that bacteria will bind more to an adhesive region downstream versus upstream on the lower surface of a channel (Figure 7a). While this occurs due to sedimentation, this phenomenon was not described by prior studies of the impact of sedimentation on uniform surfaces [63]. In addition, bacterial attachment to lower but heterogeneous surface areas decreases when flow increases (Figure 7b). This appears to be counterintuitive because flow is predicted to increase mass transport that depends on advection and diffusion [65,67] and, for uniformly adhesive surfaces, to have little effect on sedimentation-dependent mass transport [45]. These findings have implications for bacterial adhesion both in vivo and in vitro, as long as the compartment through which bacteria flow is close to being horizontal, and the lower surface of the compartment is heterogeneous so that only some regions support bacterial adhesion.

In addition, the increase in concentration of suspended bacteria near the lower surface and the rate of bacterial adhesion to an adhesive spot are expected to be highly sensitive to the size of the bacteria, because adhesion increases with sedimentation velocity and decreases with diffusion coefficient (Figure 4), and larger bacteria sediment more rapidly and diffuse more slowly. Indeed, for a spherically shaped bacteria, the concentration near the lower surface will increase with the fifth power of the radius (Appendix A), which means that within a sample of bacterial cells of heterogeneous size (which can happen even if bacteria belong to the same species and are grown under the same conditions, as we observed by the wide range of sedimentation velocities we measured in Appendix A), the larger bacteria will be transported much more rapidly to the lower surface. Our concepts may help distinguish transport effects that depend on the geometry of the compartment from more fundamental impacts of bacterial size on adhesion due to differences in the number of receptors, curvature, or drag forces.

Among other implications, our findings provide insight into studies of shear-dependent bacterial adhesion to spots of immobilized receptors. Many such studies show that bacterial attachment decreases with increased shear rates [29,57,78]. Our findings suggest that this could, at least in part, reflect shear-inhibited transport due to reduced sedimentation (Figure 4c, last panel) rather than shear-inhibited intrinsic attachment rates. At the same time, some other studies have shown that bacterial attachment to spots of receptors increases with shear rates [29,55,70], and our findings here demonstrate that this cannot be explained by transport. Instead, shear-enhanced attachment to an adhesive spot must require a specific mechanism of mechanical activation, such as activation of catch bonds [60] or the presence of drag forces pressing bacteria closer to the surface to engage more receptors [79]. Notably, our conclusions are limited to heterogeneous surfaces, in which the adhesive spot does not cover the entire lower surface of the chamber, and the field of view is near the upstream edge of the spot. Most studies of bacterial adhesion to biomaterials, as well as some studies of adhesion to immobilized receptors, use uniform surfaces [63]. In such studies, the phenomenon we report here would not apply, since it requires that the bacteria sediment form a dense layer near the lower surface while fluid passes over the non-adhesive upstream region of the chamber.

Our findings also suggest that, for in vitro studies, it is important to use identical geometry and adhesive spot locations when comparing the surface interactions of different bacteria in flow, e.g., of multiple bacterial strains to the same receptors [29,57,78], or the same strain to multiple receptors [55]. Unfortunately, this is not possible if multiple spots of different receptors are placed on the same surface to increase the efficiency of measurements, such as in microarrays. In this case, our findings show that the total adhesion to a spot will be highly sensitive to the position of the spot, and even to the position of other adhesive spots (Figure 4c and Figure 5b). Possible solutions suggested by our study would be to use transport calculations to normalize the data, or to design microarrays to distribute replicate spots across the chamber. A more tractable solution might be to characterize the pattern of adhesion within each spot as an additional metric. A high relative density of bacteria at the upstream edge of the adhesive spot would indicate high adhesivity, while a more uniform distribution of bacteria over the spot would indicate weak adhesivity, as seen in Figure 6c. This approach should be robust because the pattern of adhesion was seen to be independent of position (Figure 4c, Appendix A) and upstream adhesive spots (Figure 5b).

Finally, our findings may be relevant to bacterial infections in vivo. Indeed, smaller arteries, most veins, kidneys or other urinary tract compartments, tear or salivary ducts, indwelling catheters, etc., have laminar flow. While some such compartments run vertically, so that gravity would not bring bacteria preferentially to any side, depending on the body position, some will run horizontally or at a relatively horizontal angle at least for a certain period of time, so that bacteria will settle towards one surface, as we describe here. Moreover, such compartments generally present a heterogeneous surface for bacterial binding, which is a first step in the development of infection due to the formation of bacterial biofilms or cell invasion. Bacteria that are introduced into the body compartments with horizontal surfaces would be predicted to concentrate in downstream regions of the infected compartment. In situations such as these, the localization of biofilms or focal intracellular invasion may largely reflect the physics of bacterial transport, instead of or in addition to other molecular and/or physiological factors like the differences in surface or cellular biochemistry, or the location at which bacteria were introduced to the fluid.

## 5. Conclusions

We draw the following conclusions about bacterial binding in flow to heterogeneous surfaces. First, bacterial adhesion to an adhesive region located on the lower downstream surface of a flow compartment is much higher than to an adhesive region upstream, on an upper surface, or to a uniformly adhesive surface. This is because bacteria concentrate on the lower surface due to sedimentation while flowing over the non-adhesive regions of the surface. Second, initial bacterial attachment to an adhesive region of a heterogeneous surface may be inhibited by shear due to mass transport effects alone because higher shear washes out the sedimented bacteria. Third, to compare the adhesive strength for bacteria binding to two spots on different locations of a surface in flow, it is possible to determine when differences reflect intrinsic binding strength instead of mass transport by normalizing binding based on position, or by characterizing how sharply adhesion drops off with distance within a spot.

## Figures and Tables

**Figure 1 pathogens-12-00941-f001:**
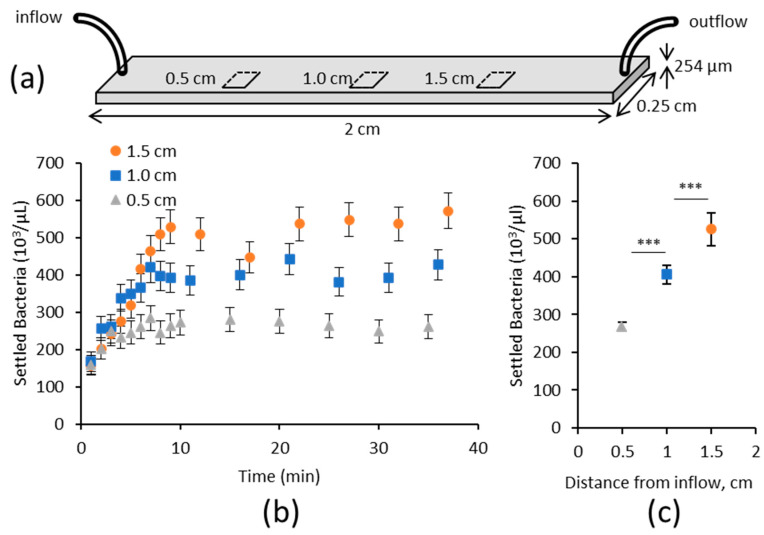
Concentration of bacteria within 2.5 microns of a nonadhesive lower surface at a wall shear rate of 4.7 s^−1^. (**a**). Schematic of the Glycotech flow chamber, showing the location at which images were taken near the lower surface of the chamber. (**b**) The concentration was determined by dividing the number of bacteria in the image by the area in the field of view and a depth of 2.5 microns. Time-dependent changes in this near-surface concentration as measured experimentally at the indicated distances downstream from the inflow of the chamber. Error bars indicate the 67% confidence interval for a Poisson distribution based on the number of bacteria per field of view. (**c**). The near-surface concentration of bacteria at equilibrium as a function of location, based on the average measurements between 10 and 40 min. *** indicates *p* < 0.0001 by one-way ANOVA with Tukey post hoc correction for multiple comparisons.

**Figure 2 pathogens-12-00941-f002:**
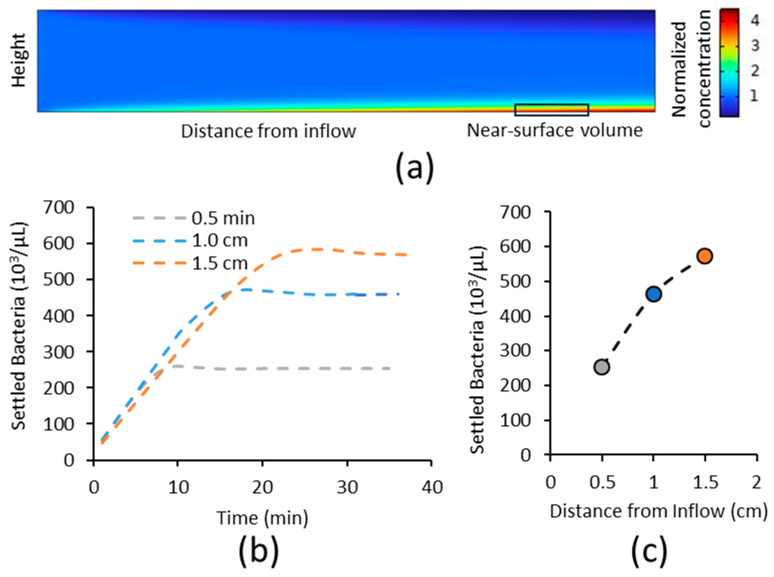
Simulations of the concentration of bacteria flowing within 2.5 microns from the lower surface. (**a**) Illustrative simulation of bacterial concentration in a chamber, viewed from the side. The small rectangle outlined in black illustrates the side view of a 2.5 µm high “near surface” volume over which the bacterial concentration is averaged to compare with experimental measurements. (**b**) Time-dependent changes in near-surface concentration at three locations within the chamber. (**c**) Equilibrium values of near-surface concentration as a function of distance from the inflow. The same colors and symbols are used in panels (**a**,**b**) to indicate distance into the channel. Parameters were used from Table 1, except that the shear rate was S = 4.7 s^−1^, and the bulk concentration was C0 = 30,000/µL.

**Figure 3 pathogens-12-00941-f003:**
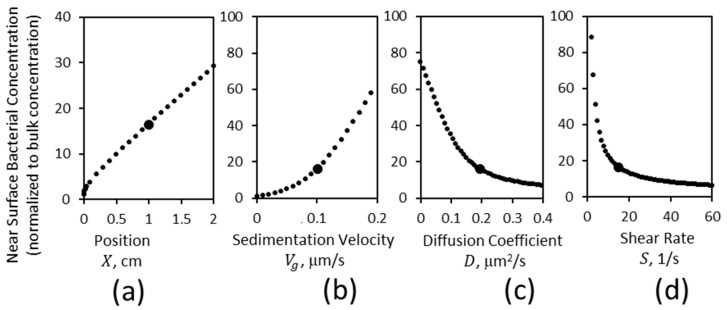
Effect of position and experimental conditions on the near-surface concentration of bacteria at the lower surface. Transport simulations were used to predict the effect of (**a**) position in distance from inflow, (**b**) sedimentation velocity, (**c**) diffusion coefficient, and (**d**) shear rate on the concentration of bacteria within 2.5 microns of the lower surface at equilibrium. All parameters are taken from Table 1 for the large circles, and only the indicated parameter varies for the small circles.

**Figure 4 pathogens-12-00941-f004:**
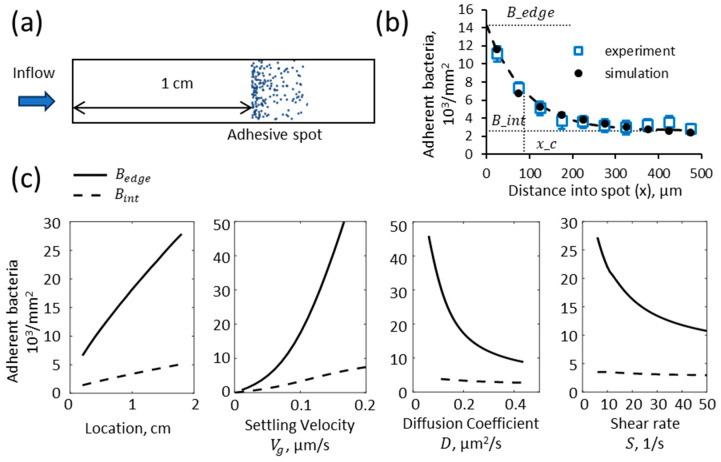
Density of adherent bacteria. (**a**) Schematic of the experiment showing bacteria (blue) adhering in an adhesive spot (not to scale). (**b**) Density of bound *E. coli* in 50 µm wide strips, after adhering for 29 min at a wall shear rate of 15 s^−1^ (average and SEM of N = 4 videos (Appendix A). Simulations were fit to the data by adjusting $k_{on}$, the association rate constant. The values of $k_{on}$ and other parameters are given in Table 1. The exponential Equation (7) is fit to the data, and the dotted lines show the meaning of the upstream edge density $B_{edge}$, the interior density $B_{int}$, and the fall-off distance $x_{c}$. (**c**) Simulations were then performed to vary one parameter at a time, as shown on the horizontal axis, with all other parameters shown in Table 1, and upstream edge density ($B_{edge}$ ), interior density ($B_{int}$ ) and fall-off distance ($x_{c}$ ) determined by fitting Equation (7) to the bound bacteria. The fall-off distances are shown in Appendix A.

**Figure 5 pathogens-12-00941-f005:**
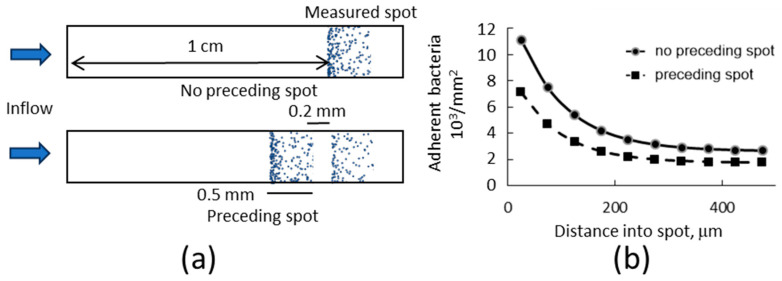
Effect of surface heterogeneity on downstream measurements. (**a**) Cartoon (not to scale) of the simulated conditions. An adhesive spot was placed 1 cm from the inflow to which bacterial density was measured after 29 min, either with a preceding spot or with no preceding spot. The spot was 0.5 mm wide and was placed 0.2 mm upstream. (**b**) Adherent bacterial density in 50 µm wide strips within the measured spot. All parameters were taken from Table 1.

**Figure 6 pathogens-12-00941-f006:**
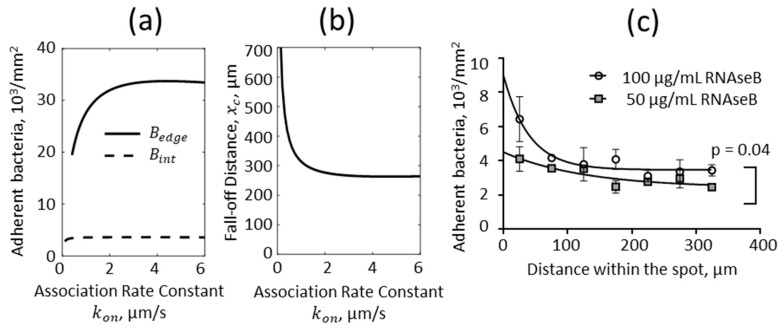
Impact of the effective association rate constant on bacterial binding. The effective association rate constant $k_{on}\;$was varied in computational simulations, and the pattern of bound bacteria was characterized as before by the (**a**) upstream edge density ($B_{edge}$), interior density ($B_{int}$), and (**b**) fall-off distance ($x_{c}$). (**c**) In experiments, adhesive spots were created 1 cm into the chamber, by incubation of the lower surface with droplets with 100 µg/mL RNaseB, versus with 50 µg/mL RNaseB and 50 µg/mL BSA, to reduce the immobilized RNAseB receptor by approximately two-fold. Bacteria were then washed over the surface at a wall shear rate of 15 1/s, and the density of bacteria was measured at 29 min. Each data point represents the mean and standard deviation of n = 2 experiments, with n = 6 images analyzed on each day (Appendix A). The *p*-value that the two concentrations of RNAseB yielded different levels of bacterial adhesion was calculated using a two-way ANOVA. The lines are Equation (7) fit to the data, with the parameters discussed in the text.

**Figure 7 pathogens-12-00941-f007:**
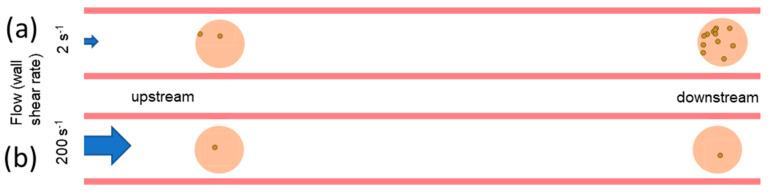
Illustration of the impact of position and fluid flow on bacterial adhesion to a heterogeneous lower surface. (**a**) At a low shear rate, binding to an adhesive (peach) region of the lower surface increases with the distance the fluid travels over a nonadhesive (white) lower surface. (**b**) At high shear stress, the amount of binding to an adhesive (peach) region of the lower surface increases very little with distance from the inflow, so binding appears shear-inhibited at downstream spots.

**Table 1 pathogens-12-00941-t001:** Parameters used in simulations, unless otherwise indicated.

Symbol	Definition	Default Value
** $V_{g}$ **	Gravitational sedimentation velocity ^1^	$0.1\;\frac{\mu m}{s}$
** *D* **	Bacterial diffusion coefficient ^1^	$0.19\;\frac{{\mu m}^{2}}{s}$
** *r* **	Bacterial radius	1 µm
** $k_{on}$ **	Effective bacterial association rate ^2^	$0.36\;\frac{\mu m}{s}$
** $C_{0}$ **	Initial bacterial concentration ^2^	$13,200\;\frac{1}{\mu L}$
** *H* **	Chamber depth ^3^	254 µm
** *S* **	Wall shear rate ^3^	$15\frac{1}{s}$
** $x_{0}$ **	Location of upstream edge of spot ^3^	1 cm

^1^ See methods. ^2^ Fit to data. ^3^ Controlled by experimental conditions.

## Data Availability

The experimental bacterial adhesion raw data are presented as Appendix A. The computational models are not publicly available because the simulations were performed on COMSOL software, so the models do not stand alone, and the key analysis of those data are presented in the manuscripts. The models will be made available by the corresponding author upon request.

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
