# Peer review of "Effect of Gravity on Bacterial Adhesion to Heterogeneous Surfaces"

_pathogens, 2023, doi:10.3390/pathogens12070941_

Round 1

Reviewer 1 Report

Hogan et al. present a very interesting manuscript investigating the adhesion of bacteria to heterogeneous surfaces under shear stress. By heterogeneous surface, the authors mean a surface with both non-adhesive and adhesive regions. They examined the effects of sedimentation, shear rate and diffusion on the adhesion of bacteria in their model system, a flow chamber with adhesive spots on an otherwise non-adhesive surface. They find that bacteria tend to concentrate on the lower surface due to sedimentation, and this results in a higher number of adherent bacteria at the upstream edge of adhesive spots. This has implications for how bacteria would adhere in in vivo settings, where receptors may not be evenly distributed. They bolster their experimental work with computational modelling. I am not in the best position to comment on the validity of the physics behind the models, but from my rather limited perspective this seems to have been carried out robustly and convincingly. I found the study to be well written and presented, and I only have relatively minor comments to improve it.

1. The authors state that they cannot share their simulation code. While this may be the case for technical reasons, I see no reason why they could not provide their microscopy datasets by depositing them in e.g. FigShare. I would urge the authors to do so.

2. The manuscript is generally well written, though there is a tendency to use different notations for units. For instance, in some cases seconds are given using the SI standard ’s’, but other times seconds are notated as ’sec’. Please stick to ’s’ throughout.

3. In the materials & methods, the authors do a fairly good job of describing their experimental setup. However, for readers unfamiliar with the flow cell systems used, the authors could provide a schematic and/or photo of their system in the supplement to help readers understand it. The same applies for the chamber in which they measure the sedimentation velocity.

4. Lines 65 and 403: this is a rather nitpicky comment, but ’convection’ formally refers to flow due to a temperature gradient. I think the term the authors should use here for mechanical flow is ’advection’.

5. Line 97: well done to the authors for using the correct name of lysogeny broth! They could mention which particular formulation they used (Lennox or Miller, or something else?).

6. Lines 101-102: please provide a reference for RNAseB being highly mannosylated.

7. Lines 206-207: t-tests are not appropriate here due to multiple comparisons. ANOVA should be used.

8. Figures 2-6: these figures contain odd lines and boxes, at least in the pdf version I am viewing. In figure 6c, the scale for the x-axis is not visible. Please make sure the final images are free of these artefacts.

9. Lines 275 and 374: 29 minutes is given, which seems an oddly specific time. This is fine, I am just wondering why not 25 or 30 minutes?

10. Equation 1: this is not the first equation in the manuscript; the previous equations should be numbered as well.

11. Line 302: the k in kon should be in italics, and the on should be a subscript.

12. Line 353: please remove the extra period.

13. Lines 357 and 359: The authors discuss a ’binding constant’. I assume they mean the association rate constant? There is a danger of confusion here, as a ’binding constant’ could be thought to refer to an equilibrium association or dissociation constant. Please be specific.

Author Response

we realized the formatting was messed up on our pasted response, so please also see pdf attached, under response to reviewer 1.

Reviewer 2 Report

This paper describes bacterial adhesion in microfluidic chambers as influenced by physical parameters such as sedimentation velocity, diffusion coefficient, surface adhesivity. It discusses the implication of such influences on bacterial adhesion in the environment of human tissue, particularly during pathogenesis. The relevance of this study has been sufficiently elaborated, the experiments are well designed and results interpreted reasonably. Comments of the following points would enrich the discussion and conclusion sections:

1. Indeed, many arteries, veins and urinary tract compartments run horizontal for at least a small extent. However, they do run vertical or at an angle predominantly. The effect of the factors considered in this study would be very different under such a circumstance.

2. Would it be possible to simulate the effect of an adhesive area that contains adhesins of differing/ mixed affinities? If yes, comments on the possible outcomes would be interesting.

3. As they stand the figures are very informative. However, micrographs, time lapse photographs or videos of adhesion would make it easier to visualize the correlation of the simulation with actual experimentation. Consider adding these in supplementary material.

Minor improvements to language clarity could be made (e.g., line 63 and line 89) which would improve the read.

Author Response

please see attachment, under response to reviewer 2.

Reviewer 3 Report

Dear authors,

the manuscript is very interesting and should be published.

Some minor comments are:

- line 15 and others, also line 183: it would be more comprehensible for the reader to write "lower surface" instead of "surface" (search for "surface" in the document)

- line 49 - 65: there seem to be missing some references?

- in Materials and Methods it would be fine to have a picture of the flow chamber, maybe also a picture taken by the Nikon camera and the phase contrast microscope to have a better idea what could be seen

- line 130-131: is the velocity always higher higher up?

- Figure 7: it should be added what is meant by 2/s and 200/s, otherwise the figure is not understandable without reading the whole text

Author Response

please see attached, under response to reviewer 3
